ecology/genetics

comparative phylogeography, freshwater fishes, genetic variation, Last Glacial Maximum, idiosyncratic responses

**Authors for correspondence:**
P. F. Victoriano
e-mail: pvictori@udec.cl
C. P. Muñoz-Ramírez
e-mail: carlos.munoz@umce.cl

# Contrasting evolutionary responses in two co-distributed species of *Galaxias* (Pisces, Galaxiidae) in a river from the glaciated range in Southern Chile

P. F. Victoriano[1], C. P. Muñoz-Ramírez[2,3,4], C. B. Canales-Aguirre[5,6], A. Jara[6], I. Vera-Escalona[3,4], T. Burgos-Careaga[1], C. Muñoz-Mendoza[6] and E. M. Habit[7]

[1]Departamento de Zoología, Facultad de Ciencias Naturales y Oceanográficas, Universidad de Concepción, Concepción, Chile
[2]Instituto de Entomología, Universidad Metropolitana de Ciencias de la Educación, Santiago, Chile
[3]Departamento de Ecología, Facultad de Ciencias, Universidad Católica de la Santísima Concepción, Concepción, Chile
[4]Centro de Investigación en Biodiversidad y Ambientes Sustentables (CIBAS), Universidad Católica de la Santísima Concepción, Concepción, Chile
[5]Centro i~mar, Universidad de Los Lagos, Camino Chinquihue Km 7, Puerto Montt, Chile
[6]Núcleo Milenio de Salmónidos Invasores (INVASAL), Concepción, Chile
[7]Unidad de Sistemas Acuáticos, Centro de Ciencias Ambientales Eula-Chile, Universidad de Concepción, Concepcion, Chile

PFV, 0000-0001-5403-3404; CPM-R, 0000-0003-1348-5476; CBC-A, 0000-0002-8468-6139; IV-E, 0000-0002-2452-6694; EMH, 0000-0002-5113-5496

Life-history traits are among the most important factors affecting population abundance and genetic diversity of species. Here, we analysed the genetic patterns of two *Galaxias* species with different life-history traits to investigate how these biological differences impacted their evolution in the Valdivia River basin, Southern Chile. We analysed mitochondrial DNA (mtDNA) sequences from 225 individuals of *Galaxias maculatus* and 136 of *G. platei* to compare patterns of genetic diversity, structure and demographic growth across the basin. *Galaxias maculatus* presented higher genetic diversity and higher genetic structure than *G. platei*. Demographic analyses showed *G. maculatus* kept a higher population size over time, with a signal of

demographic expansion in the last 250 kyr. Whereas *Galaxias platei*, exhibited lower, but constant population size over time. Furthermore, haplotype networks revealed higher lineage diversity in *G. maculatus* with a tendency to occupy different areas of the basin. Coalescent simulations ruled out that genetic differences between species could be explained by stochastic processes (genetic drift), suggesting species-specific biological differences as responsible for the observed genetic differences. We discuss how differences in life-history traits and past glaciations interact to shape the evolutionary history of the two *Galaxias* species.

## 1. Introduction

Phylogeographic studies in freshwater fish have recently increased our understanding on the relevant processes shaping biodiversity in southern South America [1,2]. Most studies have acknowledged the importance of the Andean orogeny and the Last Glacial Maximum (LGM) as drivers of general and conspicuous phylogeographic patterns such as strong trans-Andean divergence [3–6] and a trend of lower levels of genetic diversity toward more climatically unstable areas at higher latitudes [4,7]. However, some studies have also revealed striking phylogeographic patterns that have suggested less straightforward scenarios [8–10]. For instance, Zemlak *et al*. [9] found close relationships between some geographically close (i.e. similar latitude) trans-Andean populations of *Galaxias platei*, suggesting a complex Patagonian history of drainage translocations across the Andes, triggered by transitions between glacial and interglacial periods. It is not yet clear though how strong or consistent these patterns are across taxa, as freshwater phylogeography in Patagonian systems is still in an early stage. Moreover, comparative studies of co-distributed species are still rare. Some of these studies, such as Ruzzante *et al*. [8], have emphasized the relevance of comparing patterns across species to understand how biological differences may influence species' demographic responses to shared historical events.

Although few, comparative phylogeographic studies on Patagonian freshwater fish have suggested that species can be differently impacted by similar historical events at both large [8] and small [11] geographical scales. Ruzzante *et al*. [8] found that the cold-water species *Galaxias platei*, which currently inhabits areas [4,5] that were directly impacted by the Quaternary glaciations, experienced strong bottlenecks in recent glaciations, whereas the (partially) co-distributed, but more warm-adapted *Percichthys trucha* was able to maintain more stable populations across the last two glacial periods, possibly due to their ability to survive in refugia away from areas of high glacial impact. At a much smaller geographical scale (i.e. within a single river basin), but studying a larger number of species, Victoriano *et al*. [11] found a wide range of genetic patterns suggesting these should represent different demographic responses to shared historical events due to species-specific biological traits. Thus, species with certain characteristics (e.g. low habitat specificity or high salinity tolerance) were able to maintain stable populations and high genetic diversity throughout glacial cycles, while others struggled with bottlenecks and low genetic diversity. However, these studies are scarce. Furthermore, most species in these studies are phylogenetically distant, which complicates interpretations about the relation between biological traits and genetic patterns as the number of traits exhibiting variation generally increases with phylogenetic distance [12]. Indeed, in the study of Victoriano *et al*. [11], only a few species shared a close relative within the same genus (*Galaxias maculatus* with *G. platei* and *Aplochiton taeniatus* with *A. zebra*). Nevertheless, *G. maculatus* and *G. platei*, both being widespread and abundant within the Valdivia River basin, could represent an excellent opportunity to study the effects of biology and history on demographic species' responses in greater detail.

The genus *Galaxias* encompasses several cool-temperate adapted species distributed across the former regions of Gondwana [13]. *Galaxias maculatus* (Jenyns 1842) and *G. platei* (Steindachner 1898) are freshwater fish that coexist across the Valdivia River basin [14–16]. Despite being congeneric species, they display interesting biological differences making them good candidates for a comparative analysis. *Galaxias maculatus* is a facultative amphidromous species [13] which can be found in marine coastal waters, estuaries, rivers and lakes [17–19]. This species is widely distributed in Chile and Argentina [17,20], and is one of the freshwater species with the widest distribution range worldwide, in accordance to the pan-austral Gondwanian distribution proposed for the Galaxiidae family [13]. The distribution of this species in South America starts at about 34°S and extends south all over Patagonia. *Galaxias platei*, on the other hand, is restricted to South America and Falkland Islands, where it shows a distribution pattern similar to *G. maculatus*, although more southernward, starting at about 38°S in Chile and 41°S in Argentina down to Tierra del Fuego. In contrast to *G. maculatus*, *G. platei* is a strictly freshwater species and reaches much larger body sizes (350 versus 150 mm in *G. maculatus*) [21]. It can be found in lakes and rivers and presents

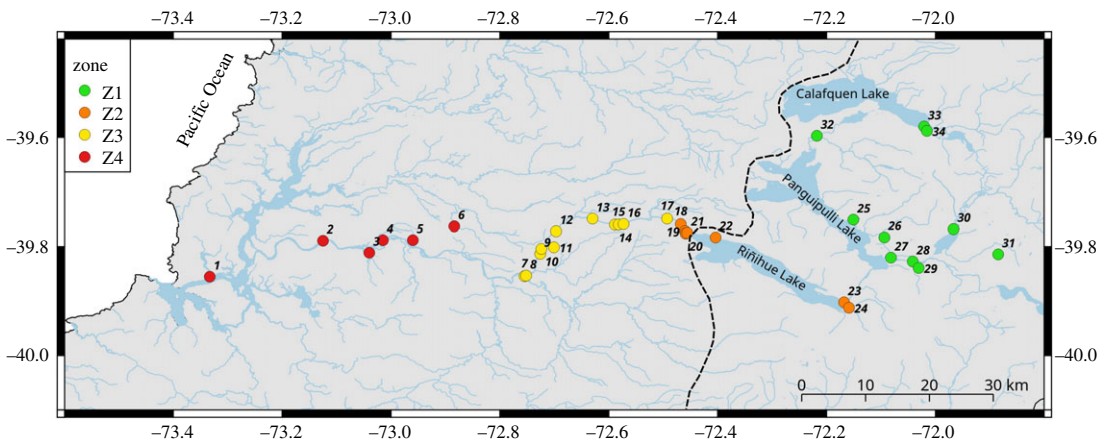

**Figure 1.** Distribution map, showing sampling localities and river zones in the Valdivia River basin, South Chile. Dotted line represents the west margin of the ice sheet during the Last Glacial Maximum.

physiological adaptations such as tolerance to the abrasion of the branchial epithelium and high mitochondrial complexity in the retinal cones [22,23], which have been suggested as adaptations to the stressful conditions of turbid glacial lakes. Due to these differences, *G. platei* appears to have persisted throughout glacial maxima in local refugia at high elevation [5,6,24,25]. By contrast, *G. maculatus* retains numerous populations living in potamal, estuarine and coastal areas that were less affected by glaciations [26].

In the last few years, a number of phylogeographic studies have contributed to the knowledge of the evolutionary history of southern South American *Galaxias* [5–10,25]. Genetic studies in *G. maculatus* have provided regional-level genetic information to understand broad-scale evolutionary processes. For instance, Zemlak *et al*. [26] analysed the genetic patterns of mtDNA across of *G. maculatus* finding that the current distribution of this species most likely reflects a west–east colonization process, followed by localized dispersal events across the Andes via drainage reversal during glacial–interglacial shifts. González-Wevar *et al*. [7] conducted a deeper study across the southern Chilean coast, finding surprisingly high levels of genetic structure in *G. maculatus*. Furthermore, this study also showed the Valdivia basin harbouring an endemic lineage, suggesting the strong isolation of its populations. Unfortunately, the study only included estuarine samples. Recently, Delgado *et al*. [16] have revealed genomic differentiation between estuarine and resident populations of *G. maculatus*, including populations of the Valdivia River basin. Regarding *G. platei*, much lower genetic structure and diversity have been reported in the Valdivia basin [11] when using mtDNA, although higher genetic diversity has been found when analysing microsatellite markers [6].

However, differences in observed patterns of genetic diversity between co-distributed species can be due not only to differences in their biological traits but also to random processes like coalescent stochasticity. Thus, a strong statement about the role of biological differences in the generation of disparate genetic patterns should minimally account for random genetic processes like genetic drift [27]. Whether the differences in genetic patterns between *Galaxias* species are due to their biological differences or random variations due to coalescent stochasticity is a question that remains untested, yet approachable with appropriate, model-based methods. In this study, we analyse a large number of populations and individuals using mtDNA sequences of both *G. maculatus* and *G. platei* with the aim to understand in greater detail their patterns of genetic diversity across the Valdivia River basin. Our specific goals are (i) to thoroughly document the genetic diversity and structure across the Valdivia River basin for *G. maculatus* and *G. platei*, (ii) to infer their demographic histories in relation to past glacial events, and (iii) to evaluate whether differences in the genetic patterns observed between species (if any) are greater than expected under a null model of random variation due to genetic drift.

## 2. Material and methods

### 2.1. Study area and sampling localities

The Valdivia River basin (40° S) is a complex Andean basin with two headwaters, located in east (Lake Lácar, Argentina) and west (Lake Pellaifa, Chile) Andes (figure 1). Both headwaters converge at Lake

Panguipulli and later in Lake Riñihue where San Pedro River begins, finishing in the Valdivia River where it finally reaches the Pacific Ocean.

A total of 225 *G. maculatus* individuals and 136 *G. platei* individuals were collected from 39 locations in four zones across the Valdivia River basin (electronic supplementary material, table S1). These four adjacent zones encompassing roughly similar areas differ in particular characteristics such as the type of environment (lacustrine or fluvial), the orientation of the subwatersheds, and the type of river stretch (hypo-rhithron/potamon). These zones correspond to: Zone 1 (Z1), lentic bodies and watercourses draining north–south and west–east, from Lake Calafqén to Lake Panguipulli; Zone 2 (Z2), watercourses and lakes draining west–east, from the eastern edge of Lake Riñihue onwards, including its tributaries and outlet (39°46′ S, 72°02′ W); Zone 3 (Z3), hyporitral type watercourses, spanning from the mouth of Lake Riñihue to the confluence of the Rivers San Pedro and Quinchilca (39°51′ S, 72°45′ W); and Zone 4 (Z4), composed mostly of potamal stretches of river, from Quinchilca to the Cutipay bridge on the Calle Calle River (39°51′ S, 73°19′ W) (figure 1). All individuals were fixed in ethanol (96%) and stored at −20°C for further molecular laboratory procedures.

## 2.2. Laboratory procedures

Total DNA was extracted from muscle tissue, using the QIAGEN DNeasy Tissue DNA extraction kit following the manufacturer's instructions. The control region fragment was amplified using the kit GoTaq Green Mastermix Promega® where each reaction was conducted in a total volume of 30 µl, containing 1X PCR buffer, 2.5 mM $MgCl_2$, 0.5 mM of primer L19 [28], 0.5 mM of primer 12SarH [29], 0.02 mM dNTPs, 0.025 U µl$^{-1}$ GoTaq DNA Polymerase® (Promega®). PCR amplification was performed according to the following programme: a cycle of 95°C for 5 min, 60°C for 30 s and 72°C for 2 min, followed by 40 cycles of 95°C for 30 s, 54°C for 45 s and 72°C for 2 min, and a final elongation cycle of 72°C for 5 min. Each PCR product was purified using Millipore filter plates and sequenced in the DNA Sequencing Center of the Department of Integrative Biology at Brigham Young University (EEUU), and Macrogen® sequencing service. All sequences were visually inspected, reviewed, edited and aligned using the program CodonCode Aligner (CodonCode Corporation, Dedham, MA). Sequences were deposited in GenBank database under the accession numbers MT339714–MT340074.

## 2.3. Genetic diversity and population structure

For each species, we estimated the haplotype number (K), haplotypic diversity (Hd), number of segregating sites (S) and nucleotide diversity (π) as indexes of genetic diversity. These indexes were estimated with DNASP v. 5.0 software [30]. Levels of among-population genetic differentiation were estimated by pairwise $F_{ST}$ and hierarchical analyses of molecular variance (AMOVA) in ARLEQUIN v. 3.5 [31] for each species. We conducted an AMOVA in order to quantify genetic variation among individuals within sites ($F_{ST}$), among sites within groups ($F_{SC}$) and among groups ($F_{CT}$). Significance of F-statistics was achieved using 10 000 permutations. We tested with AMOVA different combination of zones as hypotheses (i.e. (Z1)(Z2)(Z3)(Z4); (Z1)(Z2 + Z3 + Z4); (Z1 + Z2)(Z3 + Z4); (Z1 + Z2 + Z3)(Z4)).

## 2.4. Phylogeographic and historical demographic analyses

Demographic analyses were carried out using three approaches: (i) the neutrality tests Tajima's D [32] and Fu's Fs [33], where we estimated deviations from equilibrium based on 1000 coalescent simulations in DNASP, and we expected to yield negative values when a population faces a process of sudden population expansion; (ii) a genealogical relationship among haplotypes by a network, where we represented the haplotype frequencies by zone, the number of mutational steps among haplotypes, and the relationships among these haplotypes for each species. The network analyses were constructed using the media-joining algorithm [34] implemented in NETWORK v. 4.610 software (Fluxus Engineering); and (iii) a Bayesian skyline plot (BSP) based on coalescent theory, where we inferred the historical changes in effective population size ($N_e$) through time [35] for each species. We conducted these analyses in BEAST v. 1.7 software [36]. Each analysis was carried on to generate a posteriori distribution of effective population size employing the Markov chain Monte Carlo sampling method (MCMC). The distributions obtained were used to generate credible intervals that represent phylogenetic and coalescent uncertainty [35]. Before running BEAST, we estimated the best evolutionary model fitted to each database by species using the Akaike criteria implemented in JMODELTEST v. 2.0 software [37]. The best model for each species was HKY + G, which was incorporated for further analyses in BEAST.

Before carrying out the BSP run, we identified which clock model implemented in BEAST best fitted to our dataset (i.e. strict clock model, lognormal relaxed clock model, exponential relaxed clock model and random local clock model) [35,36,38]. We conducted preliminaries running in BEAST using default parameters and comparing likelihoods of each clock model using Bayes factor (BF) implemented in TRACER v. 1.6, which measures the weight of the evidence in the proposed model against another candidate model [39]. The best clock model that fitted our dataset was the uncorrelated lognormal relaxed clock model [35] for both species, *G. platei* and *G. maculatus* (BF > 2). Finally, we ran 10 independent analyses of MCMC with a length chain of 10 000 000, sampled each 500th iteration, and we discarded 10% of initial samples given its low estimated likelihood in the first iterations. Frequencies of each base were estimated empirically, we set the HKY + G model, and a substitution rate (μ) of 0.01876 substitutions/site/Myr [26,40]. The same parameters were used for the dataset of both species, *G. platei* and *G. maculatus*.

## 2.5. Testing whether genetic differences are beyond coalescent stochasticity

We conducted genetic simulations to test whether any dissimilarity found between the genetic patterns of the two studied species are likely under a purely coalescent stochastic process, rather than other biological or demographic explanations. For this, we used the empirical data from one species to parametrize a model of a single population, and then we used that model to see whether the genetic patterns of the other species can be reproduced at some degree. If patterns are reproducible in at least 5% of the simulations, we then would infer that the genetic patterns between species are not statistically different. All genetic simulations were conducted with SIMCOAL 2.0 [41]. We first performed 1000 simulations of a single population of *G. platei*, sampling 72 mtDNA sequences of 786 bp, and assuming a population size of 10 000 individuals. We varied the mutation rate randomly, selecting values between 0.00000001 and 0.000001 from a uniform distribution. For each of the simulations, we calculated the number of segregating sites (S) with a command line version of the software ARLEQUIN. We then used this simulated data to select a mutation rate that was able to produce the empirical S value of *G. platei* (S = 34). Instead of using the value suggested by the linear regression between mutation rate and S, we used a more conservative approach by using the 5% lower quantile regression to select the highest possible mutation rate value that would still produce at least 5% of simulated S values equal or higher than 34 (electronic supplementary material, figure S2). We then used this mutation rate to simulate 10 000 replicates of a population of *G. maculatus*, using the same population size as for *G. platei*, but sampling 217 sequences of 505 bp (i.e. same number of sequences and sequence length as in the empirical dataset of *G. maculatus*). We calculated for every simulated replicate the number of segregating sites (S), the number of haplotypes (K), and the average number of nucleotide differences (π) using ARLEQUIN. Finally, we compared the distribution of these simulated summary statistics with the empirical values obtained for *G. maculatus* to estimate the proportion of simulations that are equal to or higher than the empirical values. Failing to reproduce the observed patterns of *G. maculatus* in at least 5% of the simulations would be strong evidence that differences between these two species are not due to coalescent stochasticity.

# 3. Results

## 3.1. Genetic diversity and population structure

Overall, genetic diversity and structure were lower in the freshwater strict *G. platei* than in the diadromous *G. maculatus*. The genetic diversity indexes for *G. platei* showed a total of 44 haplotypes (K), 56 segregating sites, a haplotypic diversity (Hd) of 0.96 and a value of nucleotide diversity (π) of 0.0033 (table 1). Segregated by river zones, the number of haplotypes (K) ranged from 6 to 22, segregating sites (S) ranged from 9 to 29, the haplotypic diversity (Hd) ranged from 0.837 to 0.908, and π ranged from 0.0026 to 0.0032 (table 1). For *G. maculatus*, there were a total of 160 haplotypes, 110 segregating sites, a haplotype diversity of 0.946 and a value of π of 0.0274 (table 1). Specifically, the number of haplotypes ranged from 33 to 71, the segregating sites ranged from 66 to 89, the haplotypic diversity 0.872 to 0.993, and the nucleotide diversity ranged from 0.0255 to 0.0363 (table 1).

Pairwise $F_{ST}$ values between zones for *G. platei* ranged from 0.0466 to 0.1748 (table 2) with Zone 1 and Zone 2 being the only zones differentiated significantly, although with a very low value ($F_{ST}$ = 0.0466; $p < 0.05$). Conversely, for *G. maculatus*, all pairwise $F_{ST}$ values showed significant differences, with $F_{ST}$ values ranging from 0.031 to 0.567 (table 2). The hierarchical analyses of molecular variance

**Table 1.** Summary statistics for genetic diversity in *G. maculatus* and *G. platei*. n, sample size; S, number of segregating sites, K, Number of haplotypes, Hd, haplotype diversity, $\pi$, nucleotide diversity.

|  | zones | n | S | K | Hd | $\pi$ |
|---|---|---|---|---|---|---|
| *G. maculatus* | Zone 1 | 77 | 87 | 71 | 0.993 | 0.0255 |
|  | Zone 2 | 42 | 66 | 35 | 0.973 | 0.0349 |
|  | Zone 3 | 56 | 77 | 33 | 0.872 | 0.0277 |
|  | Zone 4 | 50 | 89 | 40 | 0.940 | 0.0363 |
|  | basin | 225 | 110 | 160 | 0.946 | 0.0274 |
| *G. platei* | Zone 1 | 46 | 21 | 22 | 0.908 | 0.0027 |
|  | Zone 2 | 49 | 29 | 20 | 0.851 | 0.0026 |
|  | Zone 3 | 31 | 28 | 11 | 0.837 | 0.0028 |
|  | Zone 4 | 10 | 9 | 6 | 0.889 | 0.0032 |
|  | basin | 136 | 56 | 44 | 0.960 | 0.0033 |

**Table 2.** Gene flow (above diagonal) and $F_{st}$ values (below diagonal) for *Galaxias maculatus* and *G. platei*.

|  | Zone 1 | Zone 2 | Zone 3 | Zone 4 |
|---|---|---|---|---|
| *G. maculatus* |  |  |  |  |
| Zone 1 | — | 15.4600 | 1.0000 | 0.3800 |
| Zone 2 | 0.0313[a] | — | 2.3300 | 0.5600 |
| Zone 3 | 0.3318[a] | 0.1468[a] | — | 1.2300 |
| Zone 4 | 0.5678[a] | 0.4700[a] | 0.2880[a] | — |
| *G. platei* |  |  |  |  |
| Zone1 | — | 10.2200 | 2.3600 | 3.4700 |
| Zone 2 | 0.0466[a] | — | 539.9000 | ∞ |
| Zone 3 | 0.1748 | 0.0009 | — | ∞ |
| Zone 4 | 0.1261 | −0.0384 | −0.0040 | — |

[a]Significant values.

**Table 3.** Hierarchical analysis of molecular variance (d.f.) for *Galaxias maculatus* (G. m.) and *G. platei* (G. p.) from the Valdivia River.

| groups | among group | | among populations within groups | | within groups | |
|---|---|---|---|---|---|---|
|  | G. m. | G. p. | G. m. | G. p. | G. m. | G. p. |
| 1 (Z1)(Z2)(Z3)(Z4) | 24.62[a] | 3.24 | 22.81[a] | 6.49[a] | 52.56[a] | 90.27[a] |
| 2 (Z1)(Z2 + Z3 + Z4) | 12.83 | 7.21 | 35.20[a] | 5.18 | 51.98[a] | 87.61[a] |
| 3 (Z1 + Z2)(Z3 + Z4) | 30.13[a] | 7.42[a] | 22.84[a] | 5.22 | 47.02[a] | 87.36[a] |
| 4 (Z1 + Z2 + Z3)(Z4) | 33.97[a] | −1.15 | 23.91[a] | 8.87[a] | 42.12[a] | 92.27[a] |

[a]Significant values.

(AMOVA) for *G. platei* showed significant differences among groups only for the comparison (Z1 + Z2) (Z3 + Z4), with 7.42% of variance explained (table 3). Regardless of how zones were compared, nearly all variance was explained by differences within populations, ranging from 87.4% to 92.3% of the variance observed (table 3). Conversely, for *G. maculatus*, all zone arrangements tested produced

relatively high and significant values (table 3). The arrangement (Z1 + Z2 + Z3)(Z4) produced the highest value of variance explained with a 33.97% (table 3).

## 3.2. Phylogeography and historical demographic analyses

Demographic analyses based on the neutrality test Tajima's D showed a negative value for *G. platei* (D = −1.570) and a positive value for *G. maculatus* (D = 0.886), with both estimations being non-significant ( $p > 0.05$ ). Conversely, for Fu's Fs, estimations for both species were negative and significant (*G. maculatus* Fs = −23.65, $p = 0.01$; *G. platei* Fs = −21.66, $p < 0.001$ ). Mismatch analyses showed a unimodal pattern for *G. platei*, consistent with scenarios of demographic instability and recent bottlenecks, whereas *G. maculatus* showed a strongly multimodal pattern, frequently associated with a more stable demographic history (electronic supplementary material, figure S1).

Regarding haplotype networks, species produced highly contrasting patterns. Genealogical relationships for *G. platei* showed some relatively abundant and geographically widespread haplotypes connected with several singletons separated by few mutational steps (figure 2*a*). For *G. maculatus*, on the contrary, three highly divergent haplogroups were identified (figure 2*b*). The haplogroup I shows a star-like network, mainly represented by individuals from zone 3 and zone 4 and a few individuals from zone 1, and it is separated by 25 mutational steps from the haplogroup II (figure 2*b*). The haplogroup II shows one main haplotype shared mainly by zones 2 and 3, with a few individuals from zone 1. This haplogroup is separated by 23 mutational steps from the haplogroup III (figure 2*a*). The haplogroup III shows many low-frequency haplotypes from all zones, although zones 1 and 2 were represented with the highest frequency. Only a few haplotypes were shared between two or more zones probably because of the low frequency of haplotypes (figure 2*b*).

The demographic reconstruction via the skyline plot analysis for *G. maculatus* showed an increase in $N_e$ that took place about 300 000 years ago (figure 3*a*). Conversely, for *G. platei*, $N_e$ showed a constant population size over time and much lower $N_e$ values compared with *G. maculatus* (figure 3*b*).

## 3.3. Genetic differences are too large to be due to coalescent stochasticity

We conducted genetic simulations to test whether the genetic patterns found for both *G. platei* and *G. maculatus* were likely to arise due to coalescent stochasticity as the sole explanation. Failure to reject the null hypothesis would preclude the need for invoking deterministic explanations such as differences in life-history traits (e.g. dispersal capabilities). Our results show that the difference in genetic diversity patterns observed in *G. platei* and *G. maculatus* are greater than those expected under coalescent stochasticity only (figure 4*a*–*c*). The number of segregating sites simulated for the hypothetical species with *G. platei* characteristics ranged from 17 to 102 with a mean of 44.76, while the empirical value for *G. maculatus* was 116 (figure 4*a*). In the case of the number of haplotypes, the simulated values ranged from 13 to 46 with a mean of 26.8, whereas the observed value for *G. maculatus* was 152, a value three times larger than the mean of the simulated values (figure 4*b*). Finally, for the average number of nucleotide differences, the simulated values ranged from 0.6 to 39.9, whereas the empirical value for *G. maculatus* was 25.1 (figure 4*c*). Here, only 20 out of 10 000 values were larger than the empirical value for *G. maculatus* (25.1), indicating a very low probability of obtaining values equal to or larger than the empirical value for *G. maculatus* ( $p$ -value = 0.002). These results indicate that the strong genetic differences observed between species are not due to coalescent stochasticity.

# 4. Discussion

We found strong differences in the patterns of genetic diversity and structure between *Galaxias maculatus* and *Galaxias platei* in the Valdivia River basin. *G. maculatus* was characterized by populations displaying generally high genetic diversity (i.e. K and Hd), higher genetic structure, and increase in $N_e$ throughout time. On the other hand, *G. platei* appears to have not being seriously affected by environmental changes of the past, presenting lower but constant population size. Variation in genetic patterns across co-distributed species could be due to differences in biological traits (i.e. that differentially impact demographic responses) or to stochastic processes such as mutational or coalescent stochasticity [42]. Although differences in genetic patterns between species can be expected at some degree due to random processes, our simulation analyses and results rule out this explanation for the species

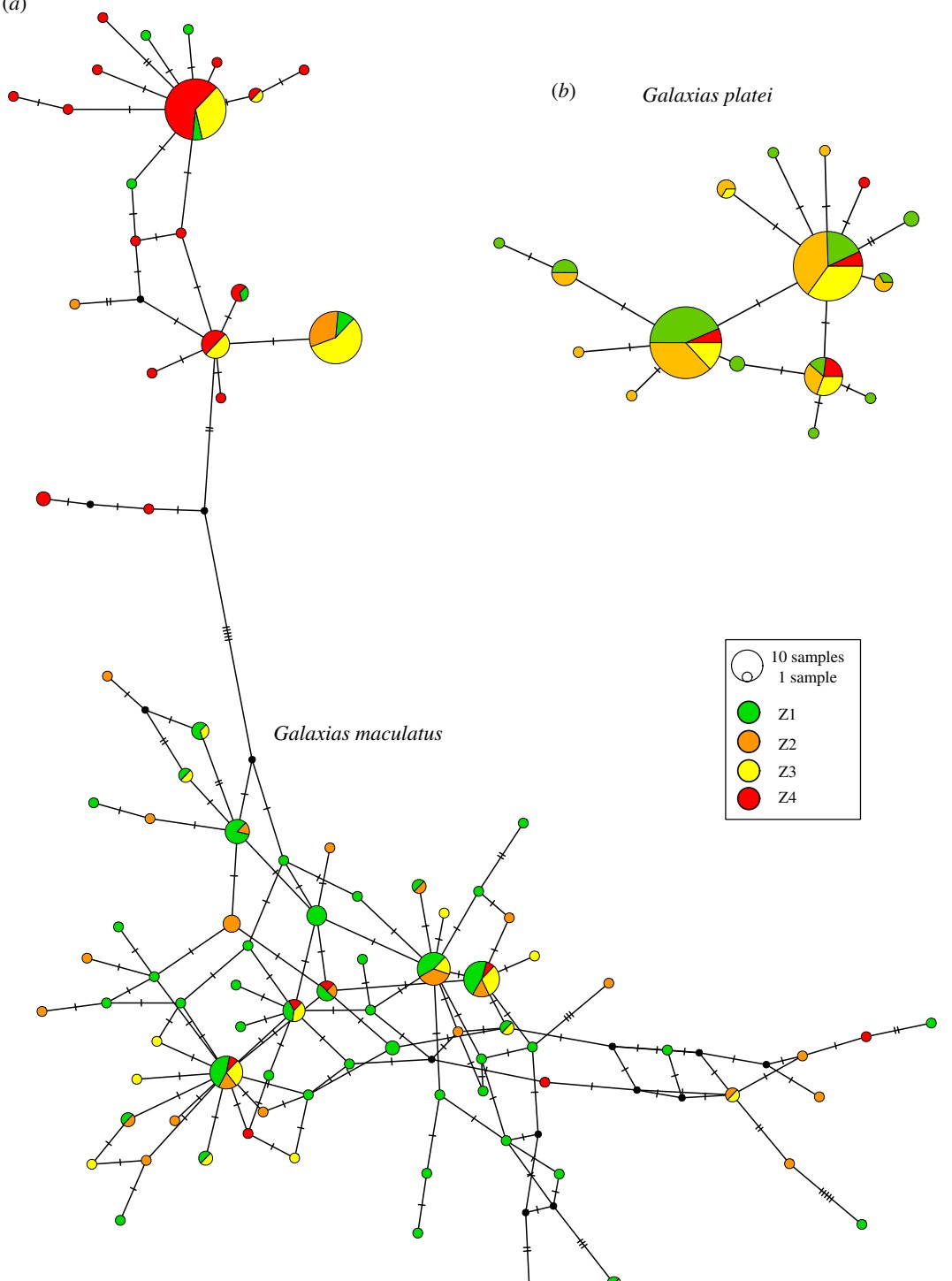

**Figure 2.** Haplotype network (median-joining) for (*a*) *Galaxias maculatus* and (*b*) *Galaxias platei* based on the mtDNA D-loop region. Circle size is proportional to haplotype frequency; Colours represent the study zones as represented in the distribution map (figure 1).

analysed here, strongly suggesting the role of biological differences as drivers of differential demographic and genetic responses in *G. maculatus* and *G. platei*.

## 4.1. Disparate genetic patterns in relation to biological differences

Idiosyncratic responses in terms of genetic diversity have been observed in several other studies, highlighting the relevance of biological attributes of each taxon in predicting microevolutionary

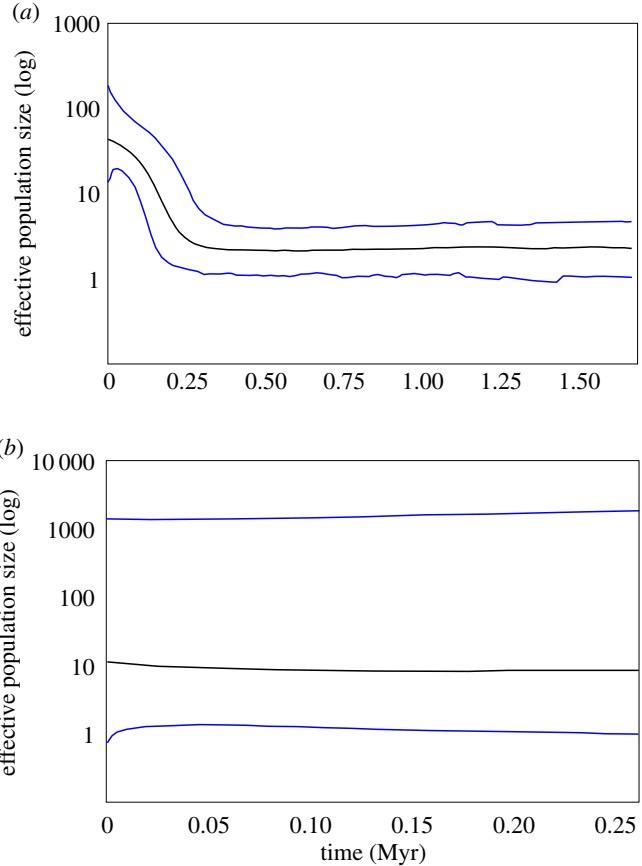

**Figure 3.** Historical estimates of female effective population size through time constructed using the Bayesian skyline model for (*a*) *Galaxias maculatus* and (*b*) *G. platei* from the Valdivia River basin based on the mtDNA D-loop region. Myr, million years.

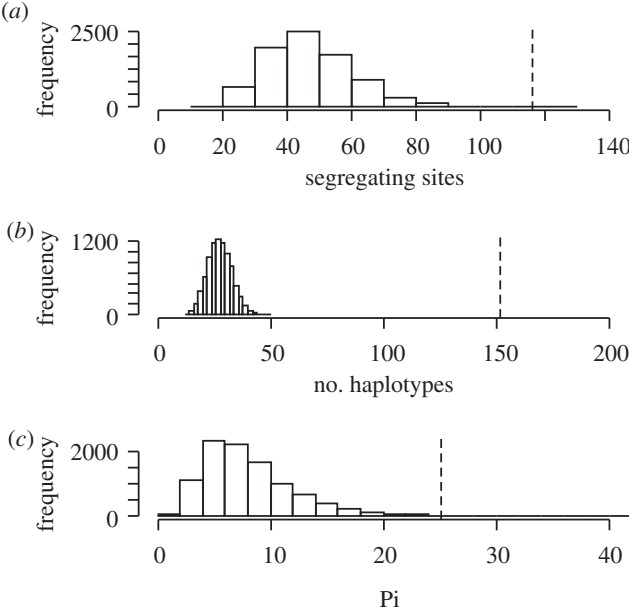

**Figure 4.** Results from the genetic simulation analyses. Genetic data were simulated under parameters that fitted the empirical data of one species (*Galaxias platei*) to test whether differences in genetic patterns in relation to the other species (*G. maculatus*) were possible under coalescent stochasticity as the only process acting to produce variation. Frequency distributions for three summary statistics—distribution for segregating sites (*a*), number of haplotypes (*b*), and pairwise number of nucleotide differences (*c*)—obtained from the simulated data are compared against the empirical values of *G. maculatus* (vertical dashed line).

responses [8,43,44]. The migratory ability of G. maculatus and its tolerance to the marine environment could have provided the species with a good chance to survive glacial periods in lower freshwater reaches of the river. By contrast, G. platei may have had to face a harsher environment, with more difficulties to maintain stable populations during colder periods due to, perhaps, its lack of active migratory behaviour [45]. These biological differences could have played a key role in shaping distinct patterns of genetic diversity. On the one hand, the migratory ability of G. maculatus might allow dispersal across larger distributional areas, connecting more distant populations, escaping local extinctions and maintaining larger population sizes (as in a metapopulation [46]). Consequently, larger populations will maintain high levels of genetic diversity by reducing the impact of genetic drift. On the other hand, the lack of a migratory behaviour in G. platei, would impact on its ability to maintain large populations and re-colonize local environments after glaciations. This should impact on the levels of intra-basin genetic diversity and increase the risk of local extinctions. Nevertheless, the adaptations of G. platei to colder environments [22,23] may have allowed the survival of some populations in situ (within the basin), reducing the risk of local extinction, albeit with lower genetic diversity. Alternatively, differences in genetic diversity could also be influenced, at least in part, by the different placement of the Valdivia basin within the overall distribution of both species. For G. maculatus, the Valdivia basin is placed in a more intermediate position of its distribution, giving the chance for a greater genetic interchange with neighbouring basins from north and south. By contrast, the G. platei population from the Valdivia basin is at the north margin of the species distribution, allowing genetic interchange only with southern populations or, via river reversals, from Atlantic basins [9].

The pattern of little change in $N_e$ and its low value found for G. platei, based on the Bayesian skyline plot results, is consistent with the idea of a persistent and strong drift effect driven by survival in less suitable environments. It is not clear if the current populations of G. platei from the glaciated range have an origin in downstream refuge areas either in the Pacific or Atlantic basins, or could represent remnants from populations that persisted within the glacier range. Although evidence of the latter have been found for some taxa like crustaceans [47], amphibians [48], fish [49], beetles and plants [50] (see also [51], for a review), Vera-Escalona et al. [6] found that survival in refugia near the coast and East-Andes areas was the most supported hypothesis for Pacific populations of G. platei. For G. maculatus, the finding of significant genetic structure and the presence of more than one lineage may violate the panmixia assumption of the Bayesian skyline plot analysis. Hence, Bayesian skyline plot results for G. maculatus should be interpreted with caution.

In addition to genetic diversity, G. maculatus also exhibited higher genetic structure, which could suggest that gene flow between some populations is currently restricted. Physical barriers to gene flow are not evident in the Valdivia River basin, so gene flow should more likely be restricted by non-physical (e.g. ecological) barriers. One possibility could be that genetic structure is being driven by population differences in life-history traits (e.g. diadromous versus resident populations). Campos [52] highlighted the occurrence of migratory diadromous populations in G. maculatus from the lower Valdivia River basin, but little had been discussed of populations with other strategies in the basin. However, Górski et al. [19], who analysed isotopic data for G. maculatus across several basins including the Valdivia River basin, showed evidence of the existence of upstream populations completing their life cycle entirely in the freshwater environment [10] and supported later with more genetic evidence [16]. Similar patterns have also been documented for Argentinean populations [53]. Differing life-cycle strategies between upstream and downstream populations could result in reproductive isolation, and therefore limited gene flow, if these differences imply that reproduction takes place in different zones of the basin (i.e. upstream versus downstream) and/or at different times of the year (different months), reducing encounters between individuals of these different populations. Our data are consistent with this scenario as one haplogroup was mostly frequent upstream, while the other two were more frequent downstream (figure 2a). This spatial variation in life-cycle strategy has been observed in G. maculatus from other areas (like in some Argentinean basins), where the species can exhibit landlocked populations in interior lakes [18,53,54].

Molecular structure driven by variation in migratory strategies have been documented in other fish species, suggesting this process can play a key role in driving genetic structure and ecological speciation [55–58]. In addition, the maintenance of these lineages represented by different ecophysiological variants (distinct phenotypes) within a single population has been suggested as a potential strategy to face selective pressures within heterogeneous and unpredictable environments. This regulation mechanism of conditional migratory plasticity, evolutionary stable strategy (ESS), would be sustained by genetic variation that allows for the evolution of migratory behaviour, as it has been proposed for eels [59]. On the other hand, differences in migratory strategies have been shown to be important for between-

basin genetic structure. Evidence of greater genetic differences for non-diadromous versus diadromous *Galaxias* species have been documented in New Zealand [60] as well as for several other fish species, where comparisons of genetic structure among freshwater, marine and anadromous species have shown greater structure between freshwater species [62].

Alternatively, deep divergences between currently sympatric haplogroups could also be explained by long-term isolation in the past followed by a relatively recent secondary contact. Given the glacial history of Southern Chile, this could be possible if different populations were isolated in different refugia during glaciations, which would later join the Valdivia River basin from these different origins. However, this explanation would require some level of current reproductive isolation between populations in order to preserve the integrity of different haplogroups within a single basin; otherwise, as a panmictic population, its lineages would be progressively sorted by genetic drift over long periods of time. Furthermore, the relatively high genetic structure observed between high and low areas of the basin supports the idea that these populations possess ecological differences that limit gene flow. Nevertheless, the historical isolation and secondary contact counts with evidence in previous studies. Geomorphological work [62] have evidenced drainage reversals, causing that some Andean lakes discharging to the Atlantic in the past (during glaciations), shifted to the Pacific after glaciations. These reversals would have facilitated dispersal of freshwater species across the Andes, from Atlantic to Pacific basins, as supported by genetic studies on fish [4,9,26,63]. This hypothesis has been supported for *G. platei* [9] and *G. maculatus* [26] and suggests that it might well be the case for some (but not all) the haplogroups found in our study.

The potential confluence of individuals from distinct lineages and geographical origins to reproduce in a single basin (philopatry of multiple lineages) could also explain the presence of multiple lineages in the Valdivia River basin, as it has been shown for *G. platei* from other basins [6]. Delgado *et al.* [16] have recently found that estuarine populations maintain high levels of gene flow between neighbour basins, while resident populations present more limited inter-basin movements. However, this explanation seems unlikely for some groups of *G. maculatus* like the one studied by González-Wevar *et al.* [7] from offshore the Valdivia River mouth as this population was genetically distinct from all other populations from neighbouring basins, suggesting this lineage was unique and endemic. Adding the samples from [7] to our samples to build a neighbour joining tree revealed this offshore population is indeed unique, being also absent from all other localities upstream of the Valdivia Basin (see electronic supplementary material, figure S3), suggesting diversity in *G. maculatus* is far higher and warranting a more comprehensive study.

In summary, the microevolutionary responses that can be displayed by fish when faced with environmental and historical factors are species specific, given the biological singularity of each taxon [64]. We show how different genetic patterns found between *G. maculatus* and *G. platei* are consistent with their biological differences. The ability of *G. maculatus* to evolve different migratory strategies (e.g. amphidromy, residency) may have provided the ability to escape glacial impacts in refugia near the coast, allowing the species to maintain larger effective population size, which in turn, preserved high levels of genetic diversity avoiding the eroding effect of strong genetic drift. However, whether these migratory strategies have the potential for promoting and maintaining multiple lineages within the Valdivia River basin is an aspect that remains to be further investigated and is beyond the scope of the present study. The absence of an active migratory behaviour in *G. platei*, could explain its lower population size and thus, lower genetic diversity, and its adaptations to cold environments could additionally explain the persistence of the species in a highly impacted basin during glaciations. Our work shows how biological differences between species cannot only impact on their current patterns of genetic diversity but also influence different evolutionary responses to historical events.

Ethics. This study was approved by the ethics committee of the Universidad de Concepcion, Chile. Permit for fishing was provided by Subsecretaria de Pesca y Acuicultura, Ministerio de Economia, Fomento y Turismo, Chile (Res. Ex. 1042).

Data accessibility. DNA sequences: GenBank accession numbers MT339714–MT340074.

Authors' contributions. P.F.V. and E.M.H. designed the sampling design, analysed data and drafted the manuscript. C.M.-M. obtained the molecular data and conducted molecular analyses. A.J. collected the samples and proof-read the manuscript. C.B.C.-A., I.V.-E., T.B.-C. and C.P.M.-R. analysed the data, conducted molecular analyses, wrote sections of the initial draft and copy-edited the manuscript.

Competing interests. The authors declare no competing interests.

Funding. This study was funded by Colbún S.A., and the additional support from the following research grant nos. FONDECYT 1161650 (P.F.V.), FONDECYT 11180897 (C.B.C.-A.) and FONDECYT 3180331 (C.P.M.-R.).

Acknowledgements. We acknowledge the collaboration of Roberto Cifuentes, Jorge González, Alejandra Oyanedel, Germán Montoya, Néstor Ortiz, Katherin Solis and Beatriz Muñoz in the collection of biological samples; we acknowledge the help of Mariem Dib and Bárbara Inzunza with laboratory work.

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
