## [Reviewer comments · Royal Society Open Science]

Review History

RSOS-200632.R0 (Original submission)

Review form: Reviewer 1

Is the manuscript scientifically sound in its present form?

No

Are the interpretations and conclusions justified by the results?

No

Is the language acceptable?

No

Do you have any ethical concerns with this paper?

No

Have you any concerns about statistical analyses in this paper?

No

Recommendation?

Accept with minor revision (please list in comments)

Comments to the Author(s)

The authors have addressed an interesting, but difficult to unravel question around species specific differences in population genetic characteristics. The paper is clear and easy to follow, although there are some very minor issues with English that could be improved.

I think the authors have done a good job discussing a range of potential explanations for the genetic patterns observed, but there is one that I think the historical biogeographic factors needs clearer recognition, although it is slightly alluded to near the very end. Rio Valdivia is the northern most *G. platei* population in Chile, thus it is at the extreme range end for the species which has implications for historical gene flow in this basin from surrounding areas. In contrast this river is broadly speaking closer to the middle of the range of *G. maculatus*, thus allowing for greater historical biogeographic connections from all directions, which may partly explain the presence of multiple very divergent lineages present in this river. To me, just knowing the differences in their geographic distributions would lead to the prediction of differences irrespective of any other factor. That's not to say other factors don't matter, but this one is likely to be important.

This point leads into my next one, the authors mention the previous study of *G. maculatus* (reference 10, but several others as well) using the same mtDNA region which found multiple major different lineages, one of which was endemic to the Valdivia Basin (indeed, there are several studies looking at broader patterns). Do the lineages found in the current study match those from the previous study? Or are there additional new lineages? If they are the same as those found previously then this suggests that many Chilean rivers are likely to have a mix of lineages present and that the simple north south pattern of haplotypes found in the earlier study was driven by limited sampling. If the lineages appear to be limited to the Valdivia Basin this would be quite significant as it would imply some local factors have been very important in explaining the current genetic diversity. The issue of how Valdivia *G. maculatus* populations fits into the broader range of *G. maculatus* is important given the crazy levels of genetic diversity found. It is less so for *G. platei* as the species as a whole is less diverse (although it would be good to know how Valdivia diversity fits into the broader picture with *G. platei*). Ignoring these broader patterns simplifies the manuscript, but reduces its comprehensiveness.

To follow up on this point, near the end of the manuscript the authors write "It remains to be tested whether this population may represent another population or species that has been overlooked in most genetic samplings, for which further studies are warranted." But the data are all there, all you need to do is align it all together and run a simple ML analysis! I'm not sure that referring to "this population" is very accurate since I think you are referring to the Valdivia Basin as a whole, from which you sampled 34 populations.

It would be helpful if the authors included line numbers with their manuscript.

Page 3

Line 25, former, not current regions of Gondwana

Line 40, elevation, not altitude (altitude is the height above the land surface, i.e., flying at an altitude of 30,000 feet)

Line 51, add an e to stuarine, same on page 7, line 59

Page 5

Line 13, is that divergence rate per lineage or pairwise?

Line 13 you should really be citing the original source of this calibration value, not Zemplack (26) as they simply used the rate from the Waters paper.

Page 6

Line 56, you might clarify what you mean by coastal refugia. Do you mean the lower freshwater reaches of the river, the estuary, or the ocean? Normally adult galaxiids aren't found in the ocean and mostly not in estuaries either except the upper most portion.

Line 58, is there evidence that *G. platei* lacks salinity tolerance? Most Galaxiids, even the non diadromous ones have high salinity tolerance.

Page 7

Line 12, further south there is evidence of fish persistence in glaciated areas (Unmack, P.J., Barriga, J.P., Battini, M.A., Habit, E.M. and Johnson, J.B., 2012. Phylogeography of the catfish *Hatcheria macraei* reveals a negligible role of drainage divides in structuring populations. *Molecular Ecology*, 21(4), pp.942-959) as well as other species, might be good to cite the review by Sersic, A.N., Cosacov, A., Cocucci, A.A., Johnson, L.A., Pozner, R., Avila, L.J., Sites Jr, J.W. and Morando, M., 2011. Emerging phylogeographical patterns of plants and terrestrial vertebrates from Patagonia. *Biological Journal of the Linnean Society*, 103(2), pp.475-494 plus Ashworth, A.C., Markgraf, V. and Villagran, C., 1991. Late Quaternary climatic history of the Chilean Channels based on fossil pollen and beetle analyses, with an analysis of the modern vegetation and pollen rain. *Journal of Quaternary Science*, 6(4), pp.279-291 in terms of other examples besides amphibians and crustaceans.

Line 17, The statement "cases could be indicative of low mobility or high habitat specificity" does not really conform with the fact that *G. maculatus* is one of the most widespread freshwater fish in the world which is historically extremely abundant. They mostly appear to be highly mobile and live in most habitats. Now, I know what you are trying to say, that some populations may have evolved to be less mobile or have become adapted to more specific habitats, but you need to make that point clearer. In fairness to the authors most of that paragraph backs up their point quite well, but I just think the main point could be more clearly stated at the start of the paragraph.

Line 51, Ruzzante didn't propose the geomorphological reversal of river basins! Put in some citations from the geomorphological literature. Ruzzante's work has examined evidence in the biota for these drainage reversals.

Page 8

Line 10 "The ability of *G. maculatus* to evolve different migratory strategies (e.g. amphidromy, residency) may have provided the ability to escape glacial impacts in coastal refugia and maintain larger interconnected populations, which in turn, preserved high levels of genetic diversity." But hang on, the existing evidence suggests the Valdivia lineage is endemic to this river (which could be tested by incorporating your dataset into the broader existing ones), with high population structure, thus saying they had larger interconnected populations seems to be the opposite of this statement, at least for some populations.

Figure and Table captions on page 13 have a number of minor errors such as missing italics, too many commas (Table 2 and 3, all but one should be removed), no capitals after a semi-colon, Fig 4, correct *C. maculatus*.

Very nice figures though, really like Fig 1.

Review form: Reviewer 2

Is the manuscript scientifically sound in its present form?

Yes

Are the interpretations and conclusions justified by the results?

No

Is the language acceptable?

Yes

Do you have any ethical concerns with this paper?

No

Have you any concerns about statistical analyses in this paper?

Yes

Recommendation?

Accept with minor revision (please list in comments)

Comments to the Author(s)

This paper compares mtDNA structure and diversity for 2 galaxiid species found in the Valdivia River system. The drainage is well sampled in geographic terms. As might be expected, the diadromous species shows relatively low differentiation, but higher diversity, relative to the non-migratory species. The paper is generally succinct and well written, albeit with a few typos.

The study provides a solid yet generally modest/routine contribution to the literature, with limitations surrounding the lack of replication (e.g. single river; single genetic marker; comparing a single pair of species). The main potential advance here comes from the demographic comparisons, which suggest qualitatively different trajectories for the 2 species, and give clues as to the relative effects of glaciation on them

Conclusions throughout may need to be qualified given the above limitations - e.g. as they are based on a single molecular marker (MtDNA), and unlikely to accurately reflect genome-wide patterns.

Also, the Bayesian Skyline analysis may rest on the assumption of panmixia, which may be violated given the presence of some genetic structure within the basin? This potential limitation should be acknowledged.

Additional relevant literature on meta-analyses and multispecies comparisons between freshwater and diadromous species should be cited:

Allibone and Wallis 1993 BJLS

Ward, Woodwark, Skibinski, 1994, J Fish Biol

Minor issues:

p2, line 22 - "Nevertheless, *G. maculatus* and *G. platei*, both being widespread..."

p4, l19: "encompassing roughly similar areas"

p4, l20: hiporital/potamal - words not used in english language?

p4, l49 - delete "4"

p4, l51 - demographic

p7, "impact on its ability"

Are there any geographic/geomorphological reasons to make clearer predictions regarding the distribution of genetic structure within the river system - e.g. gorges/lakes, etc? e.g. see Waters et al 2015 J Biogeography 42:2452

Decision letter (RSOS-200632.R0)

05-Jun-2020

Dear Mr Muñoz-Ramírez

On behalf of the Editors, I am pleased to inform you that your Manuscript RSOS-200632 entitled "Contrasting evolutionary responses in two codistributed species of *Galaxias* (Pisces, Galaxiidae) in a river from the glaciated range in Southern Chile" has been accepted for publication in Royal Society Open Science subject to minor revision in accordance with the referee suggestions. Please find the referees' comments at the end of this email.

The reviewers and handling editors have recommended publication, but also suggest some minor revisions to your manuscript. Therefore, I invite you to respond to the comments and revise your manuscript.

- Ethics statement

- Data accessibility

If you wish to submit your supporting data or code to Dryad (<http://datadryad.org/>), or modify your current submission to dryad, please use the following link:
<http://datadryad.org/submit?journalID=RSOS&manu=RSOS-200632>

- Competing interests

- Authors' contributions

- Acknowledgements

- Funding statement

Because the schedule for publication is very tight, it is a condition of publication that you submit the revised version of your manuscript before 14-Jun-2020. Please note that the revision deadline will expire at 00.00am on this date. If you do not think you will be able to meet this date please let me know immediately.

- 1) A text file of the manuscript (tex, txt, rtf, docx or doc), references, tables (including captions) and figure captions. Do not upload a PDF as your "Main Document";
- 2) A separate electronic file of each figure (EPS or print-quality PDF preferred (either format should be produced directly from original creation package), or original software format);
- 3) Included a 100 word media summary of your paper when requested at submission. Please ensure you have entered correct contact details (email, institution and telephone) in your user account;
- 4) Included the raw data to support the claims made in your paper. You can either include your data as electronic supplementary material or upload to a repository and include the relevant doi

within your manuscript. Make sure it is clear in your data accessibility statement how the data can be accessed;

5) All supplementary materials accompanying an accepted article will be treated as in their final form. Note that the Royal Society will neither edit nor typeset supplementary material and it will be hosted as provided. Please ensure that the supplementary material includes the paper details where possible (authors, article title, journal name).

If your manuscript is newly submitted and subsequently accepted for publication, you will be asked to pay the article processing charge, unless you request a waiver and this is approved by Royal Society Publishing. You can find out more about the charges at <https://royalsocietypublishing.org/rsos/charges>. Should you have any queries, please contact openscience@royalsociety.org.

on behalf of Dr Michael Tobler (Associate Editor)
openscience@royalsociety.org

Associate Editor Comments to Author (Dr Michael Tobler):

Associate Editor: 1

Comments to the Author:

We have received the reports from two reviewers that provided favorable reviews of the manuscript. Both reviewers provide detailed feedback that will help the authors to improve the manuscript. Once these comments are addressed, the manuscript should be suitable for publication in RSOS.

Reviewer comments to Author:

Reviewer: 1

Comments to the Author(s)

The authors have addressed an interesting, but difficult to unravel question around species specific differences in population genetic characteristics. The paper is clear and easy to follow, although there are some very minor issues with English that could be improved.

I think the authors have done a good job discussing a range of potential explanations for the genetic patterns observed, but there is one that I think the historical biogeographic factors needs clearer recognition, although it is slightly alluded to near the very end. Rio Valdivia is the northern most *G. platei* population in Chile, thus it is at the extreme range end for the species which has implications for historical gene flow in this basin from surrounding areas. In contrast this river is broadly speaking closer to the middle of the range of *G. maculatus*, thus allowing for greater historical biogeographic connections from all directions, which may partly explain the presence of multiple very divergent lineages present in this river. To me, just knowing the differences in their geographic distributions would lead to the prediction of differences irrespective of any other factor. That's not to say other factors don't matter, but this one is likely to be important.

This point leads into my next one, the authors mention the previous study of *G. maculatus* (reference 10, but several others as well) using the same mtDNA region which found multiple major different lineages, one of which was endemic to the Valdivia Basin (indeed, there are several studies looking at broader patterns). Do the lineages found in the current study match those from the previous study? Or are there additional new lineages? If they are the same as those found previously then this suggests that many Chilean rivers are likely to have a mix of lineages present and that the simple north south pattern of haplotypes found in the earlier study was driven by limited sampling. If the lineages appear to be limited to the Valdivia Basin this would be quite significant as it would imply some local factors have been very important in explaining the current genetic diversity. The issue of how Valdivia *G. maculatus* populations fits into the broader range of *G. maculatus* is important given the crazy levels of genetic diversity found. It is less so for *G. platei* as the species as a whole is less diverse (although it would be good to know how Valdivia diversity fits into the broader picture with *G. platei*). Ignoring these broader patterns simplifies the manuscript, but reduces its comprehensiveness.

To follow up on this point, near the end of the manuscript the authors write "It remains to be tested whether this population may represent another population or species that has been overlooked in most genetic samplings, for which further studies are warranted." But the data are all there, all you need to do is align it all together and run a simple ML analysis! I'm not sure that referring to "this population" is very accurate since I think you are referring to the Valdivia Basin as a whole, from which you sampled 34 populations.

It would be helpful if the authors included line numbers with their manuscript.

Page 3

Line 25, former, not current regions of Gondwana

Line 40, elevation, not altitude (altitude is the height above the land surface, i.e., flying at an altitude of 30,000 feet)

Line 51, add an e to stuarine, same on page 7, line 59

Page 5

Line 13, is that divergence rate per lineage or pairwise?

Line 13 you should really be citing the original source of this calibration value, not Zemplack (26) as they simply used the rate from the Waters paper.

Page 6

Line 56, you might clarify what you mean by coastal refugia. Do you mean the lower freshwater reaches of the river, the estuary, or the ocean? Normally adult galaxiids aren't found in the ocean and mostly not in estuaries either except the upper most portion.

Line 58, is there evidence that *G. platei* lacks salinity tolerance? Most Galaxiids, even the non diadromous ones have high salinity tolerance.

Page 7

Line 12, further south there is evidence of fish persistence in glaciated areas (Unmack, P.J., Barriga, J.P., Battini, M.A., Habit, E.M. and Johnson, J.B., 2012. Phylogeography of the catfish *Hatcheria macraei* reveals a negligible role of drainage divides in structuring populations. *Molecular Ecology*, 21(4), pp.942-959) as well as other species, might be good to cite the review by Sersic, A.N., Cosacov, A., Cocucci, A.A., Johnson, L.A., Pozner, R., Avila, L.J., Sites Jr, J.W. and Morando, M., 2011. Emerging phylogeographical patterns of plants and terrestrial vertebrates from Patagonia. *Biological Journal of the Linnean Society*, 103(2), pp.475-494 plus Ashworth, A.C., Markgraf, V. and Villagran, C., 1991. Late Quaternary climatic history of the Chilean Channels based on fossil pollen and beetle analyses, with an analysis of the modern vegetation and pollen rain. *Journal of Quaternary Science*, 6(4), pp.279-291 in terms of other examples besides amphibians and crustaceans.

Line 17, The statement “cases could be indicative of low mobility or high habitat specificity” does not really conform with the fact that *G. maculatus* is one of the most widespread freshwater fish in the world which is historically extremely abundant. They mostly appear to be highly mobile and live in most habitats. Now, I know what you are trying to say, that some populations may have evolved to be less mobile or have become adapted to more specific habitats, but you need to make that point clearer. In fairness to the authors most of that paragraph backs up their point quite well, but I just think the main point could be more clearly stated at the start of the paragraph.

Line 51, Ruzzante didn't propose the geomorphological reversal of river basins! Put in some citations from the geomorphological literature. Ruzzante's work has examined evidence in the biota for these drainage reversals.

Page 8

Line 10 “The ability of *G. maculatus* to evolve different migratory strategies (e.g. amphidromy, residency) may have provided the ability to escape glacial impacts in coastal refugia and maintain larger interconnected populations, which in turn, preserved high levels of genetic diversity.” But hang on, the existing evidence suggests the Valdivia lineage is endemic to this river (which could be tested by incorporating your dataset into the broader existing ones), with high population structure, thus saying they had larger interconnected populations seems to be the opposite of this statement, at least for some populations.

Figure and Table captions on page 13 have a number of minor errors such as missing italics, too many commas (Table 2 and 3, all but one should be removed), no capitals after a semi-colon, Fig 4, correct *C. maculatus*.

Very nice figures though, really like Fig 1.

Reviewer: 2

Comments to the Author(s)

This paper compares mtDNA structure and diversity for 2 galaxiid species found in the Valdivia River system. The drainage is well sampled in geographic terms. As might be expected, the diadromous species shows relatively low differentiation, but higher diversity, relative to the non-migratory species. The paper is generally succinct and well written, albeit with a few typos.

The study provides a solid yet generally modest/routine contribution to the literature, with limitations surrounding the lack of replication (e.g. single river; single genetic marker; comparing a single pair of species). The main potential advance here comes from the demographic comparisons, which suggest qualitatively different trajectories for the 2 species, and give clues as to the relative effects of glaciation on them

Conclusions throughout may need to be qualified given the above limitations - e.g. as they are based on a single molecular marker (MtDNA), and unlikely to accurately reflect genome-wide patterns.

Also, the Bayesian Skyline analysis may rest on the assumption of panmixia, which may be violated given the presence of some genetic structure within the basin? This potential limitation should be acknowledged.

Additional relevant literature on meta-analyses and multispecies comparisons between freshwater and diadromous species should be cited:

Allibone and Wallis 1993 BJLS
Ward, Woodwark, Skibinski, 1994, J Fish Biol

Minor issues:

p2, line 22 - "Nevertheless, *G. maculatus* and *G. platei*, both being widespread..."

p4, l19: "encompassing roughly similar areas"

p4, l20: hiporitral/potamal - words not used in english language?

p4, l49 - delete "4"

p4, l51 - demographic

p7, "impact on its ability"

Are there any geographic/geomorphological reasons to make clearer predictions regarding the distribution of genetic structure within the river system - e.g. gorges/lakes, etc? e.g. see Waters et al 2015 J Biogeography 42:2452

Author's Response to Decision Letter for (RSOS-200632.R0)

See Appendix A.

Decision letter (RSOS-200632.R1)

10-Jun-2020

Dear Mr Muñoz-Ramírez,

It is a pleasure to accept your manuscript entitled "Contrasting evolutionary responses in two codistributed species of *Galaxias* (Pisces, Galaxiidae) in a river from the glaciated range in Southern Chile" in its current form for publication in Royal Society Open Science.

You can expect to receive a proof of your article in the near future. Please contact the editorial office (openscience_proofs@royalsociety.org) and the production office (openscience@royalsociety.org) to let us know if you are likely to be away from e-mail contact -- if

you are going to be away, please nominate a co-author (if available) to manage the proofing process, and ensure they are copied into your email to the journal.

on behalf of Dr Michael Tobler (Associate Editor) and Andrew Dunn (Subject Editor)
openscience@royalsociety.org

Appendix A

Dear Editor,

We thank you and the reviewers for the comments and constructive feedback. Below you can find our point-by-point response (in blue) to all the comments and suggestions given by the reviewers which have greatly improved the content and quality of the manuscript.

Reviewer comments to Author:

Reviewer: 1

Comments to the Author(s)

The authors have addressed an interesting, but difficult to unravel question around species specific differences in population genetic characteristics. The paper is clear and easy to follow, although there are some very minor issues with English that could be improved.

I think the authors have done a good job discussing a range of potential explanations for the genetic patterns observed, but there is one that I think the historical biogeographic factors needs clearer recognition, although it is slightly alluded to near the very end. Rio Valdivia is the northern most *G. platei* population in Chile, thus it is at the extreme range end for the species which has implications for historical gene flow in this basin from surrounding areas. In contrast this river is broadly speaking closer to the middle of the range of *G. maculatus*, thus allowing for greater historical biogeographic connections from all directions, which may partly explain the presence of multiple very divergent lineages present in this river. To me, just knowing the differences in their geographic distributions would lead to the prediction of differences irrespective of any other factor. That's not to say other factors don't matter, but this one is likely to be important.

Response: Thank you for pointing this out. Although the distributional ranges are likely related to the species' biological differences we address here, we recognize that this could be a factor in explaining, at least in part, some of the striking differences in genetic patterns between species, so we have incorporated this factor in the discussion section (added towards the end of the 1st paragraph in the discussion section).

This point leads into my next one, the authors mention the previous study of *G. maculatus* (reference 10, but several others as well) using the same mtDNA region which found multiple major different lineages, one of which was endemic to the Valdivia Basin (indeed, there are several studies looking at broader patterns). Do the lineages found in the current study match those from the previous study? Or are there additional new lineages? If they are the same as those found previously then this suggests that many Chilean rivers are likely to have a mix of lineages present and that the simple north south pattern of haplotypes found in the earlier study was driven by limited sampling. If the lineages appear to be limited to the Valdivia Basin this would be quite significant as it would imply some local factors have been very important in explaining the current genetic diversity. The issue of how Valdivia *G. maculatus* populations fits into the broader range of *G. maculatus* is important given the crazy levels of genetic diversity found. It is less so for *G. platei* as the species as a whole is less diverse (although it would be good to know how Valdivia diversity fits into the broader picture with *G. platei*). Ignoring these broader patterns simplifies the manuscript, but reduces its comprehensiveness.

Response: We agree with the reviewer that this aspect would be very significant to test. However, addressing this aspect in this manuscript would shift the focus and scope of the study

significantly as it would require not only new analyses, but also changes in all major manuscript sections, including a deep treatment of *G. maculatus* biogeography.

We were aware of this point previously and we thought that such analyzes and results had implications beyond the scope of the present manuscript that needed to be covered in depth in a different manuscript which is currently in preparation. This new manuscript will cover the above-mentioned point and will expand it to incorporate analyses of phylogenetic diversity with a conservation viewpoint in mind. This is why we did not add this component to the present manuscript which focus was about the interspecific comparison between the two *Galaxias* species.

Nevertheless, recognizing the importance of this aspect we have added an additional paragraph to the discussion to rise the potential significance of a comparison with populations from other basins, also recognizing that such analysis is beyond the scope of this study.

To follow up on this point, near the end of the manuscript the authors write “It remains to be tested whether this population may represent another population or species that has been overlooked in most genetic samplings, for which further studies are warranted.” But the data are all there, all you need to do is align it all together and run a simple ML analysis! I’m not sure that referring to “this population” is very accurate since I think you are referring to the Valdivia Basin as a whole, from which you sampled 34 populations.

Response: By population we meant the marine population sampled by the previous study. This is a fair suggestion, which we have now addressed by extending the discussion at the end of paragraph 6, section 5.1. In addition, a new figure was added to the electronic supplementary material showing the relationship between samples offshore the Valdivia river mouth (from a previous study; reference [7] in the manuscript) and our samples. However, given that a manuscript exclusively about *G. maculatus* is being prepared to address the specific issue of its biogeography in South America, and that the topic goes beyond the specific question being addressed here, we decided to avoid discussing further details.

It would be helpful if the authors included line numbers with their manuscript.

Response: We worked on the template supplied by RSOS that lacked line numbers, sorry about that, although the generated pdf had them.

Page 3

Line 25, former, not current regions of Gondwana

Response: Fixed

Line 40, elevation, not altitude (altitude is the height above the land surface, i.e., flying at an altitude of 30,000 feet)

Response: Fixed.

Line 51, add an e to stuarine, same on page 7, line 59

Response: Both instances were fixed.

Page 5

Line 13, is that divergence rate per lineage or pairwise?

Response: Per lineage. Changed to the appropriate format 0.01876 substitutions/site/My

Line 13 you should really be citing the original source of this calibration value, not Zemplack (26) as they simply used the rate from the Waters paper.

Fixed: We changed this citation and corresponding reference to Waters et al. (2007) in Systematic Biology.

Page 6

Line 56, you might clarify what you mean by coastal refugia. Do you mean the lower freshwater reaches of the river, the estuary, or the ocean? Normally adult galaxiids aren't found in the ocean and mostly not in estuaries either except the upper most portion.

Response: We meant precisely that, the lower freshwater reaches of the basin. Thank you for the suggested words. Fixed.

Line 58, is there evidence that *G. platei* lacks salinity tolerance? Most Galaxiids, even the non diadromous ones have high salinity tolerance.

Response: This tolerance was assumed giving that *G. platei* is considered a resident species, but as far as we know, no evidence of lower salinity tolerance has been documented for the species. Considering that this sentence may be read as too speculative, we have removed the sentence.

Page 7

Line 12, further south there is evidence of fish persistence in glaciated areas (Unmack, P.J., Barriga, J.P., Battini, M.A., Habit, E.M. and Johnson, J.B., 2012. Phylogeography of the catfish *Hatcheria macraei* reveals a negligible role of drainage divides in structuring populations. *Molecular Ecology*, 21(4), pp.942-959) as well as other species, might be good to cite the review by Sersic, A.N., Cosacov, A., Cocucci, A.A., Johnson, L.A., Pozner, R., Avila, L.J., Sites Jr, J.W. and Morando, M., 2011. Emerging phylogeographical patterns of plants and terrestrial vertebrates from Patagonia. *Biological Journal of the Linnean Society*, 103(2), pp.475-494 plus Ashworth, A.C., Markgraf, V. and Villagran, C., 1991. Late Quaternary climatic history of the Chilean Channels based on fossil pollen and beetle analyses, with an analysis of the modern vegetation and pollen rain. *Journal of Quaternary Science*, 6(4), pp.279-291 in terms of other examples besides amphibians and crustaceans.

Response: Thanks for the suggestion. We have added this literature to the revised manuscript.

Line 17, The statement "cases could be indicative of low mobility or high habitat specificity" does not really conform with the fact that *G. maculatus* is one of the most widespread freshwater fish in the world which is historically extremely abundant. They mostly appear to be highly mobile and live in most habitats. Now, I know what you are trying to say, that some populations may have evolved to be less mobile or have become adapted to more specific habitats, but you need to make that point clearer. In fairness to the authors most of that paragraph backs up their point quite well, but I just think the main point could be more clearly stated at the start of the paragraph.

Response: Fixed. We have re-worded the beginning of the paragraph to make the point clearer and direct. Thank you for the suggestion.

Line 51, Ruzzante didn't propose the geomorphological reversal of river basins! Put in some citations from the geomorphological literature. Ruzzante's work has examined evidence in the biota for these drainage reversals.

Response: These references were changed to acknowledge the geomorphological studies, citing a recent review on the topic and referring to the references therein.

Page 8

Line 10 "The ability of *G. maculatus* to evolve different migratory strategies (e.g. amphidromy, residency) may have provided the ability to escape glacial impacts in coastal refugia and maintain larger interconnected populations, which in turn, preserved high levels of genetic diversity." But hang on, the existing evidence suggests the Valdivia lineage is endemic to this river (which could be tested by incorporating your dataset into the broader existing ones), with high population structure, thus saying they had larger interconnected populations seems to be the opposite of this statement, at least for some populations.

Response: The wording here was probably misunderstood, due to bad wording on our part. What we really meant is that the ability of the species to maintain variable life-history traits such as amphidromy, made this species to be able to thrive better in unstable or shifting environmental conditions, which in turn allowed higher survival rates, larger population sizes, and higher genetic diversity (i.e. avoiding strong genetic drift). We have changed the wording of this paragraph to make the point clearer.

Figure and Table captions on page 13 have a number of minor errors such as missing italics, too many commas (Table 2 and 3, all but one should be removed), no capitals after a semi-colon, Fig 4, correct *C. maculatus*.

Response: All these typos plus others we found throughout the manuscript were now fixed

Very nice figures though, really like Fig 1.

Reviewer: 2

Comments to the Author(s)

This paper compares mtDNA structure and diversity for 2 galaxiid species found in the Valdivia River system. The drainage is well sampled in geographic terms. As might be expected, the diadromous species shows relatively low differentiation, but higher diversity, relative to the non-migratory species. The paper is generally succinct and well written, albeit with a few typos.

The study provides a solid yet generally modest/routine contribution to the literature, with limitations surrounding the lack of replication (e.g. single river; single genetic marker; comparing a single pair of species). The main potential advance here comes from the demographic comparisons, which suggest qualitatively different trajectories for the 2 species, and give clues as to the relative effects of glaciation on them

Conclusions throughout may need to be qualified given the above limitations - e.g. as they are based on a single molecular marker (MtDNA), and unlikely to accurately reflect genome-wide patterns.

Response: The reviewer is correct and that is why we have carefully paid attention to the type of conclusions that we could make given the type of marker used, without making conclusions that could go beyond these limitations. The large sampling size of populations and individuals across the basin helped overcoming some limitations, and the use of a model-based analysis to test for genetic differences between species also contributed to this purpose.

Also, the Bayesian Skyline analysis may rest on the assumption of panmixia, which may be violated given the presence of some genetic structure within the basin? This potential limitation should be acknowledged.

Response: We have now acknowledged this potential issue in the text (see changes at the end of 2nd paragraph in section 5.1).

Additional relevant literature on meta-analyses and multispecies comparisons between freshwater and diadromous species should be cited:

Allibone and Wallis 1993 BJLS
Ward, Woodward, Skibinski, 1994, J Fish Biol

Response: Added in the context of paragraph 4, in section 5.1

Minor issues:

p2, line 22 - "Nevertheless, *G. maculatus* and *G. platei*, both being widespread..."

Response: Fixed.

p4, 119: "encompassing roughly similar areas"

Response: Fixed.

p4, 120: hiporitral/potamal - words not used in english language?

Response: Fixed.

p4, 149 - delete "4"

Response: Deleted.

p4, 151 – demographic

Response: Fixed by removing an s from demographics

p7, "impact on its ability"

Response: Fixed. Changed “in” by “on”

Are there any geographic/geomorphological reasons to make clearer predictions regarding the distribution of genetic structure within the river system - e.g. gorges/lakes, etc? e.g. see Waters et al 2015 J Biogeography 42:2452

Response: We have used the lakes as major geomorphological factors impacting genetic structure when comparing zones Z1-Z2 vs Z3-Z4 in the river. Also, zones 1-4 are defined based on other minor geomorphological features. Other than that, there are no other relevant features to make these predictions. These characteristics are defined in the 2nd paragraph from section 3.1 *Study area and sampling localities*.